# Knowledge, Attitudes, Motivations, Expectations, and Systemic Factors Regarding Antimicrobial Use Amongst Community Members Seeking Care at the Primary Healthcare Level: A Scoping Review

**DOI:** 10.3390/antibiotics14010078

**Published:** 2025-01-13

**Authors:** Nishana Ramdas, Johanna C. Meyer, Natalie Schellack, Brian Godman, Eunice Turawa, Stephen M. Campbell

**Affiliations:** 1Department of Public Health Pharmacy and Management, School of Pharmacy, Sefako Makgatho Health Sciences University, Ga-Rankuwa 0208, South Africa; hannelie.meyer@smu.ac.za (J.C.M.); stephen.campbell@smu.ac.za (S.M.C.); 2South African Vaccination and Immunisation Centre, Sefako Makgatho Health Sciences University, Ga-Rankuwa 0208, South Africa; 3Department of Pharmacology, Faculty of Health Sciences, University of Pretoria, Pretoria 0028, South Africa; natalie.schellack@up.ac.za; 4Strathclyde Institute of Pharmacy and Biomedical Sciences, University of Strathclyde, Glasgow G4 0RE, UK; 5Burden of Disease Research Unit, South African Medical Research Council, Cape Town 19070, South Africa; eunice.turawa@mrc.ac.za; 6School of Health Sciences, University of Manchester, Manchester M13 9PL, UK

**Keywords:** antimicrobial resistance, antimicrobial stewardship, primary healthcare, community members, patients, knowledge, attitudes

## Abstract

Background/Objectives: Antimicrobial resistance (AMR) is a major global health challenge, particularly in low- and middle-income countries (LMICs). Understanding the knowledge, attitudes, motivations, and expectations of community members regarding antimicrobial use is essential for effective stewardship interventions. This scoping review aimed to identify key themes relating to the critical areas regarding antimicrobial use among community members in primary healthcare (PHC), with a particular focus on LMICs. Methods: OVID Medline, PubMed, and CINAHL databases were searched using Boolean operators and Medical Subject Headings (MeSH) terms relevant to antimicrobial use and community behaviors. The Population, Intervention, Comparison, Outcome, and Study Design (PICOS) framework guided study selection, which focused on community members seeking care in PHC in LMICs. Data management and extraction were facilitated using the Covidence platform, with the Critical Appraisal Skills Programme (CASP) qualitative checklist applied for qualitative studies. A narrative synthesis identified and grouped key themes and sub-themes. Results: The search identified 497 sources, of which 59 met the inclusion criteria, with 75% of the studies conducted in outpatient primary care settings. Four key themes were identified: (1) the ’patient’ theme, highlighting beliefs, knowledge, and expectations, which was the most prominent (40.5%); (2) the ’provider’ theme, emphasizing challenges related to clinical decision-making, knowledge gaps, and adherence to guidelines; (3) the ’healthcare systems’ theme, highlighting resource limitations, lack of infrastructure, and policy constraints; and (4) the ‘intervention/uptake’ theme, emphasizing strategies to improve future antibiotic use and enhance access to and quality of healthcare. Conclusions: Stewardship programs in PHC settings in LMICs should be designed to be context-specific, community-engaged, and accessible to individuals with varying levels of understanding, involving the use of information and health literacy to effectively reduce AMR.

## 1. Introduction

Over the past two decades, antimicrobial resistance (AMR) has emerged as one of the most critical global health challenges, significantly threatening healthcare systems worldwide [1]. The recent United Nations General Assembly High-Level Meeting on Antimicrobial Resistance in September 2024 reaffirmed global commitment to addressing AMR, setting ambitious targets, including reducing AMR-associated deaths by 10% by 2030, as well as ensuring that at least 60% of countries have funded national action plans on AMR by 2030 [2]. This high-level political declaration underscores the urgency of implementing effective strategies to combat AMR, particularly in primary healthcare settings in low- and middle-income countries (LMICs), where antibiotic use in humans can constitute up to 95% of total antibiotic consumption [3].

In 2019, 4.95 million deaths were associated with bacterial AMR, with 1.27 million directly attributable to it [4]. The burden of AMR is geographically variable, with western sub-Saharan Africa experiencing the highest AMR-related death rate, at 27.3 per 100,000 people [5].

The rise of AMR presents a significant challenge by compromising the effectiveness of existing antimicrobial treatments, leaving healthcare providers with limited therapeutic options [1]. AMR has an impact not only on morbidity and mortality, but also on healthcare costs [4,6,7,8,9]. Consequently, AMR is now described as a looming pandemic unless immediate actions are taken to reverse the trend [10]. The World Health Organization (WHO) has initiated several key measures in response to this growing crisis. These include the Global Action Plan (GAP) for AMR, translating into National Action Plans (NAPs) [11,12], as well as developing the AWaRe (Access, Watch, and Reserve) list of antibiotics, with an emphasis on reducing the use of Watch and Reserve antibiotics due to their greater resistance potential [13,14]. The increasing use of Watch antibiotics is a particular issue in LMICs, adding to AMR [15,16]. Recently, the launch of the WHO AWaRe book has provided treatment guidelines for a variety of infectious diseases, aiming to enhance the appropriate use of antibiotics [17,18]. This is important in LMICs to help guide future antibiotic use away from high levels of inappropriate prescribing and dispensing of antibiotics, especially for self-limiting conditions such as upper respiratory tract infections [19,20,21,22,23]. This is because the burden of AMR is currently highest in LMICs, driven by the inappropriate use of antibiotics, including a high percentage of antibiotics from the Watch and Reserve lists [4,5,14,15,16].

Despite global efforts, AMR remains a significant problem, particularly in LMICs, including South Africa, where major AMR outbreaks have been documented [19,24,25,26]. In response, the South African government developed the Antimicrobial Resistance Strategy Framework in 2014, as well as an NAP to curb AMR [27,28]. Similarly, across Africa, NAPs have been developed to reduce current high rates of AMR. However, there are ongoing concerns surrounding available personnel and resource issues which prevent the full implementation of the activities suggested in these NAPs [28,29,30].

The key activities outlined within the NAPs include a greater understanding of current antibiotic utilization patterns across sectors and the rationale for their use, and ongoing activities to improve future utilization patterns [28,29,30]. The latter includes the instigation of antimicrobial stewardship programs (ASPs) as a critical strategy to mitigate the growing AMR threat [28,31]. ASPs typically involve coordinated interventions to promote the responsible use of antimicrobials, optimizing treatment outcomes and minimizing the spread of resistant pathogens [31,32,33]. There have been concerns surrounding the availability of the resources and personnel required to undertake these interventions in LMICs; however, this situation is changing [32,33,34,35,36].

In South Africa, the recent passage of the National Health Insurance (NHI) Act in 2024 aims to provide universal health coverage with potential implications for antibiotic stewardship and the control of AMR at the primary care level [25,33,37,38]. The NHI’s focus on equitable access to healthcare services could potentially impact prescribing practices and patient expectations regarding antibiotic use [25,38].

In LMICs, antibiotic utilization in humans typically involves both the prescribing of antibiotics in PHC centers, with nurses and other healthcare professionals (HCPs), including doctors and pharmacists, playing an appreciable role; as well as the extensive purchasing of antibiotics without a prescription from community pharmacies and drug stores [25,39,40,41,42,43,44]. Given AMR’s complexity and gravity, understanding community members’ knowledge, attitudes, motivations, and expectations regarding antimicrobial use at the PHC level is vital [45,46,47,48]. Previous research has emphasized the significant influence of patient expectations on the prescribing and dispensing of antibiotics for self-limiting conditions in LMICs, which urgently needs addressing going forward to achieve UN GA goals [2,49,50,51,52,53].

Building on Sono et al.’s (2024a) pilot studies in a rural South African province, which assessed patients’ understanding of antimicrobial use, language barriers, and reasons for self-purchasing of antibiotics [53], this review aims to further explore patient knowledge, attitudes, motivations, and awareness of AMR to inform future policies. Sono et al. (2024b) also highlighted challenges in patient comprehension of terms including “antibiotics” and “AMR”, as well as the impact of language barriers on patient education in South Africa [3,54]. This shows the importance of addressing health literacy, which has been defined by the WHO as “the ability of individuals to gain access to, understand and use information in ways which promote and maintain good health” [55]. Similarly, a recent systematic review by Wojcik et al. (2024) found comparable issues in HICs, where public campaigns, healthcare, and educational resources are more developed, underscoring that limited understanding of AMR is a global issue, rather than one confined to LMICs [56].

Despite ongoing global efforts, including national action plans and stewardship programs, critical gaps remain. Firstly, in the understanding of how community members in LMICs perceive and engage with AMR; secondly, in the role of health literacy, cultural factors, and local healthcare practices in influencing antimicrobial use; and thirdly, in the development of tailored, context-specific interventions to address these issues. Existing reviews often focus on HICs or address AMR through a narrow lens, leaving these important questions underexplored both generally and in relation to LMICs especially. This again urgently needs addressing.

This scoping review builds on existing evidence by specifically addressing these gaps. It offers a comprehensive analysis of the literature on patient knowledge, motivations, and behaviors in PHC settings in LMICs, set within the contexts of systemic factors, focusing on identifying key educational needs, cultural influences, and actionable strategies for ASPs. By considering these issues, the review aims to contribute to the development of locally relevant interventions and inform global efforts to combat AMR effectively.

It will highlight patients’ expectations of receiving antibiotics for self-limiting conditions, including colds and fevers [20,41], exacerbated by their limited understanding of antibiotics and AMR [20,57,58,59,60]. To combat AMR effectively, community-driven approaches tailored to local social and cultural contexts are essential [61]. Understanding patient perspectives is key to developing tailored interventions that address language, gender, and local beliefs and practices, as well as other cultural factors influencing antibiotic use [46,47,62,63,64,65,66,67,68].

As a result, this review aims to identify areas for education and behavior change strategies, contributing to more effective antibiotic stewardship and AMR reduction in LMICs.

## 2. Results

As shown in Figure 1, our search identified 497 records, which included 192 from CINAHL, 125 from PubMed, 114 from MEDLINE, 50 from Ovid, and a further 16 from unspecified sources. In total, 187 duplicate records were removed using the Covidence Software 2024. The titles and abstracts of the remaining 310 records were screened for relevance, with 225 records subsequently excluded for not meeting the inclusion criteria. Full-text eligibility screening was performed on the remaining 85 records, and a further 26 records were subsequently excluded. Of these, 11 had a different patient population, 7 used different study designs, 3 assessed different interventions, and 3 were excluded for different study settings. One study interviewed HCPs, and one did not report the outcomes of interest. In total, 59 articles were included in the review.

Most of the studies (73%) employed quantitative methods, while 27% used qualitative methods of assessment. Overall, Figure 1 maps out the article search outputs and the selection processes at different stages of the study process and analysis. The studies included in the analysis originated from a diverse set of countries, highlighting a broad geographical representation. Specifically, the United Kingdom (UK) accounted for the highest number of studies (15), followed by the United States (6), Spain (5), and broader European contributions (4). Canada and Australia each contributed three studies. Additional single-study contributions came from countries such as France, Brazil, Sudan, Saudi Arabia, Nigeria, Finland, Iceland, Italy, Germany, Malawi, South Africa, Singapore, China, Egypt, Norway, Myanmar, Thailand, Ireland, Malaysia, Serbia, and India. This diversity underscores the global scope and multidisciplinary nature of the research conducted.

### 2.1. Identified Themes

The 59 papers analyzed were grouped according to four key themes: (i) Patient, (ii) Provider, (iii) Healthcare System, and (iv) Intervention, Uptake, and Implementation. These four themes were further broken down into 18 sub-themes.

The 18 sub-themes were referenced a total of 116 times, indicating that individual sources often addressed multiple sub-themes (see Table 1). Specifically, 47 references (40.5%) were related to the ‘Patient’ theme, 32 references (27.6%) were connected to the ‘Provider’ theme, 21 references (18.1%) were linked to the ‘Intervention, Uptake, and Implementation’ theme, and 16 references (13.8%) were attributed to the ‘Healthcare System’ theme. Table 1 summarizes the themes and their related sub-themes, including the number of sources that they featured in.

#### 2.1.1. Patient

The Patient theme explores how various factors influence patient behavior and decision-making surrounding antibiotic use. Key findings highlight that patient expectations, beliefs, and levels of knowledge significantly impact healthcare utilization and antibiotic demand.

Trust in HCPs and effective communication play crucial roles, as do broader community attitudes and sociocultural influences. Additionally, sociodemographic factors, including educational background and health literacy, shape patients’ understanding of and approach to antibiotic use, with disparities in access and awareness evident across different populations. Overall, this theme emphasizes the complex interplay between personal, social, and cultural factors in shaping antibiotic-related behaviors.

#### 2.1.2. Provider

The Provider theme examines the critical role of HCPs in antibiotic prescribing and patient interactions. It emphasizes the dynamics of trust, communication, and decision-making between patients and providers, including how patient expectations influence prescribing practices. The findings also explore a number of other factors, including diagnostic uncertainty, adherence to guidelines, and the impact of external influences, including pharmaceutical marketing, on provider behaviors. Additionally, this theme discusses the challenges of antibiotic provision in resource-limited settings, and highlights the importance of provider and patient education for improving healthcare outcomes and addressing AMR.

#### 2.1.3. Healthcare Systems

The Healthcare Systems theme addresses the challenges and complexities involved in delivering effective healthcare, particularly in the context of AMS. It highlights resource constraints, the need for better access to healthcare services, and the impact of these factors on patient satisfaction and treatment decisions. This theme also explores the dynamics within the healthcare system, including the balance between provider dependency and patient autonomy, as well as the importance of tailoring healthcare interventions to diverse populations. Additionally, this theme covers the integration of diagnostic tools, including C-Reactive Protein (CRP) point-of-care testing. Subsequently, it discusses how technological advances influence treatment-seeking behavior and antibiotic use, while also considering the risks of increased access to healthcare advice, leading to potential overuse.

#### 2.1.4. Intervention, Uptake, and Implementation

The Intervention, Uptake, and Implementation theme explores strategies to improve antibiotic prescribing practices and enhance healthcare utilization. It examines how patients’ confidence in self-care and attitudes towards healthcare impact antibiotic use, as well as the effectiveness of various interventions, including public education campaigns, in changing behaviors and attitudes. This theme also addresses the integration of digital versus paper-based education materials, and the importance of clear communication in these initiatives. This theme also highlights shifts in knowledge, patient behaviors, and perceptions following educational efforts, while also emphasizing the role of public health messaging. Additionally, this theme explores how video and messaging strategies influence parental demand for antibiotics and reduce unnecessary antibiotic interest, demonstrating the impact of well-designed educational interventions on public understanding and attitudes.

### 2.2. Sub-Themes

The sub-theme ‘patient beliefs, expectations, knowledge and understanding, and perceptions of antibiotics, resistance, and illness severity’, categorized under the broader ‘Patient’ theme, was referenced 17 times, accounting for 15% of all references. This sub-theme highlights how patients’ understanding and misconceptions about antibiotics and AMR can influence their attitudes and behaviors. These perceptions often shape expectations of receiving antibiotics and decisions about seeking care, underscoring the importance of targeted educational interventions. This was followed by ‘uptake of interventions and program components, program effectiveness and outcomes, and public education and awareness’ which was referenced 15 times (13%), and emphasizes the challenges and successes associated with implementing AMR-related interventions. It covers how public health programs are received by communities, the factors driving their effectiveness, and the role of public education in enhancing awareness and compliance.

The third most frequently mentioned sub-theme under the ‘*Provider*’ theme was ‘*patient–provider relationship*’, referenced 13 times (11%). This sub-theme illustrates the dynamics of communication and trust between patients and healthcare providers. Evidently, effective patient–provider relationships are crucial for aligning expectations, improving shared decision-making, and addressing overprescription practices. The fourth sub-theme was ‘*healthcare system challenges and solutions, resource constraints and healthcare delivery, and healthcare system dynamics*’, which appeared 12 times (10%). This sub-theme focuses on systemic issues, such as inadequate resources, organizational inefficiencies, and broader structural barriers, that impact AMR-related outcomes. Addressing these challenges is critical for creating sustainable and equitable healthcare delivery systems. The fifth sub-theme ‘*healthcare utilization and antibiotic prescribing practices and behaviors*’, also under the *‘Provider*’ theme, was referenced 11 times (9%). This sub-theme highlights provider practices and behaviors related to antibiotic prescribing, including behaviors such as prescribing antibiotics without proper diagnostic confirmation, overprescribing due to perceived patient expectations or pressures, and sometimes prioritizing convenience or timesaving over evidence-based care.

Together, these top five sub-themes accounted for 68 references (59% of the total 116 references). The remaining 13 sub-themes were referenced 48 times (41%), collectively. Although nearly half of the sub-themes (9 out of 18) were categorized under the ‘*Patient*’ theme, only one Patient sub-theme featured among the top five most frequently referenced sub-themes, highlighting the relatively great focus on provider, system, and intervention-related factors in the included studies.

## 3. Discussion

Four key themes were identified across the 59 studies. These were ‘*Patient*’, ‘*Provider*’, ‘*Healthcare systems*’ and ‘*Intervention*, *uptake*, and *implementation*’. These themes were broken down into 18 sub-themes, providing a comprehensive understanding of the factors influencing antimicrobial use in PHC settings. Notably, the ‘*Patient*’ theme, which encompassed sub-themes related to patient beliefs, knowledge, and expectations regarding antibiotics and AMR, emerged as the most frequently referenced theme (40.5% of total references). This underscores the critical role that patient education and understanding play in driving appropriate as well as inappropriate antimicrobial use. However, targeted educational campaigns among patients can be challenging, especially if there are language barriers, particularly in multilingual or linguistically diverse populations. These issues arise when educational materials or health campaigns are not effectively translated or tailored to different linguistic groups, potentially limiting their reach and comprehension [63,123,124,125]. This further highlights the need for comprehensive research and improved surveillance systems to identify language-specific barriers and gaps in patient understanding, which can inform the development of more targeted and effective policies and interventions. Patient education emerges as a critical area to be addressed, as many studies indicated misconceptions about antibiotics, illness severity, and AMR among patients, which all contribute to the inappropriate use of antibiotics. The findings further suggest that AMS efforts should not only focus on HCPs, especially those in LMICs where the burden of AMR is greatest, but should also focus on instigating public education campaigns that are accessible and health-literate to patients and community members [25]. Similarly, a recent systematic review by Wojcik et al. (2024) reported that even in high-income countries, where public campaigns and educational resources are more advanced, challenges in terms of understanding AMR persist due to health literacy gaps [56]. This highlights that poor comprehension of AMR is a global issue, not limited to LMICs. Tailored interventions are essential for addressing health literacy levels, gender, local context, practices, and cultural factors influencing antibiotic use. The ‘*Intervention*, *uptake*, and *implementation*’ theme (18.1%) reinforces the focus on the effectiveness of public education and awareness programs, aimed at reducing inappropriate antimicrobial use. This highlights the importance of community-level interventions in tackling AMR, particularly in LMICs, where public health infrastructure may be limited. However, particular care must be taken regarding issues surrounding language in some settings [54,124].

The five most frequently referenced sub-themes, accounting for nearly 60% of the total references, provided critical insights into the most pressing issues surrounding antimicrobial use. The most referenced sub-theme was ‘*Patient beliefs, expectations, and knowledge and understanding of antibiotics*, *AMR*, and *illness severity*’, which reflects the widespread gaps in patient knowledge regarding appropriate antibiotic use. Similarly, the ‘*Uptake of interventions* and *program effectiveness*’ sub-theme emphasizes the importance of public health initiatives aimed at educating patients and communities about AMR.

The second most referenced theme was ‘*Provider*’ (27.6%), with its sub-themes highlighting the importance of patient–provider relationships and prescribing behaviors. This aligns with existing literature emphasizing the influence of HCPs on patient decision-making, particularly in terms of antibiotic prescriptions [25]. The findings suggest that improving provider–patient communication could play a pivotal role in addressing inappropriate antimicrobial use. We will be exploring this further in future research projects, now that the AWaRe book, giving treatment guidance for an appreciable number of infectious diseases, including non-antibiotic choices, has been published [17,18,126]. We are also likely to see an increase in the number of quality indicators based on the AWaRe classification and book to improve future antibiotic use, especially following the recent deliberations of the UN [2].

The ‘*patient–provider relationship*’ sub-theme, along with ‘*healthcare utilization and antibiotic prescribing practices*’, points to the significant impact that HCPs have on antimicrobial use. These findings indicate that improving provider education and training through enhancing curricula in universities and post-qualification could help to reduce unnecessary antibiotic prescribing and dispensing. We will be exploring this further in future studies involving LMICs.

The ‘*Healthcare system*’ theme (13.8%) addresses challenges including resource constraints, which often hinder the implementation of effective ASPs in low-resource settings [34,127]. However, this is now changing, with a number of ASPs being implemented in several LMICs, providing future direction [32,33,36,128,129,130]. The review findings suggest that ASPs must be tailored to address the specific needs and challenges identified in PHC settings, particularly in LMICs.

The ‘*healthcare system challenges* and *solutions*, *resource constraints,* and *healthcare delivery*’ sub-theme highlights the structural issues that exacerbate AMR in LMICs, where healthcare systems may be under-resourced and overburdened to effectively address AMR [11,19,28,29,131]. The variability in healthcare system resources, particularly in LMICs, indicates that ASPs need to be adaptable to local contexts, considering the availability of resources and the specific challenges faced by key stakeholders and healthcare systems in different regions [131,132].

This scoping review addresses the first critical gap by analyzing the ‘*Patient*’ theme, which highlights the knowledge, attitudes, and expectations of community members regarding antimicrobial use in LMICs. Findings under this theme underscore the influence of patient beliefs and misconceptions about antibiotics and AMR on healthcare-seeking behavior. For example, the frequent expectation of receiving antibiotics for self-limiting conditions reflects gaps in awareness that interventions must address. Additionally, the ‘*Provider*’ theme explores the patient–provider relationship, illustrating how trust and communication—or their absence—affect community engagement with AMR-related initiatives. Together, these themes emphasize the interplay between individual understanding and systemic interactions, highlighting the need for tailored educational interventions that engage communities as active partners in AMR stewardship efforts.

The second gap—investigating how health literacy, cultural influences, and healthcare practices shape antimicrobial use—is explored through the lens of all four key themes. The ‘*Patient*’ theme identifies disparities in health literacy and the influence of this on antibiotic demand, while the ‘*Provider*’ theme examines how healthcare practitioners navigate cultural norms and expectations when making prescribing decisions. The ‘*Healthcare Systems*’ theme highlights structural challenges, such as resource constraints and reliance on informal healthcare providers, which exacerbate inappropriate antimicrobial use. Finally, the ‘*Intervention*, *Uptake*, and *Implementation*’ theme reveals how poorly adapted public health initiatives often fail to resonate with local cultural contexts, underscoring the need for linguistically and culturally appropriate strategies. By linking these themes, this review demonstrates the complex interdependencies between individual, provider, and system-level factors which influence antimicrobial practices in LMICs.

To address the third gap, this review evaluates interventions through the ‘*Intervention*, *Uptake*, and *Implementation*’ theme, focusing on the success and limitations of existing AMR strategies. The ‘*Patient*’ and ‘*Provider*’ themes further inform the design of context-specific interventions by detailing the barriers faced by individuals and healthcare providers in adhering to appropriate antimicrobial use practices. The ‘*Healthcare System*’ theme emphasizes the need to address systemic constraints, such as diagnostic tool shortages and limited personnel, to ensure the scalability of interventions. Together, these four themes provide a comprehensive framework for understanding how interventions can be tailored to local needs, particularly in resource-limited settings. This integrative approach highlights the importance of designing interventions that are adaptable, culturally sensitive, and systemically feasible in order to effectively reduce inappropriate antimicrobial use and combat AMR in LMICs.

Most of the studies identified (75%) employed quantitative, correlational designs, which primarily explored the relationships between knowledge, attitudes, and behaviors surrounding antimicrobial use.

Furthermore, most studies (75%) were conducted in outpatient primary care settings, with a substantial proportion focusing on respiratory infections. Primary care in this review is defined as the first point of contact within the healthcare system, where individuals receive accessible and comprehensive healthcare services [133]. This encompasses care provided by general practitioners, nurses, pharmacists, and other non-specialist healthcare workers in formal settings such as clinics and community health centers, aligning with the WHO’s framework for community-based healthcare [134].

This is perhaps not surprising since, respiratory tract infections are a common condition in LMICs, and are responsible for an appreciable proportion of antibiotic use, including antibiotics purchased without a prescription [36,37,51,118,119]. These findings highlight a central focus in research on the common conditions that typically prompt antibiotic prescriptions.

The studies included in the analysis originated from a diverse set of geographical regions, highlighting the global scope and multidisciplinary nature of the research. While this review provides valuable insights into antimicrobial use and resistance across varied healthcare systems, there is a notable imbalance in representation. Most of the studies were conducted in HICs, while LMICs, which bear the highest burden of AMR, were significantly under-represented. This limits the applicability of findings to resource-constrained healthcare settings.

Despite focusing on LMICs and HICs, this scoping review emphasizes the need for context-specific research on LMICs, in particular to better understand the unique drivers of AMR and to develop tailored ASPs. Future research should address this gap by refining search strategies to capture more studies from LMICs and include non-English and gray literature. This review identified only two studies from Nigeria and South Africa, and a single study from India. This disparity underscores the significant under-representation of LMICs in research on antimicrobial use and resistance. While high-income countries provide valuable insights, the scarcity of context-specific studies in LMICs highlights a pressing need for further research in these settings. This is particularly critical to better understand the nuances of AMR and ASPs in LMIC contexts.

Overall, we believe that this review highlights several critical research gaps in the current understanding of AMR in LMICs. First, sociocultural factors such as traditional beliefs, gender roles, and health literacy remain underexplored, yet they play a pivotal role in driving antibiotic misuse. Second, the influence of informal healthcare systems, including unregulated pharmacies and community drug sellers, on AMR dynamics has been largely overlooked. We will explore this further in future studies. Lastly, there is a pressing need for interventions that are culturally and linguistically tailored to LMIC settings, as most existing strategies are modeled on high-income country frameworks that may not translate effectively to resource-constrained environments. Addressing these gaps is vital for developing sustainable solutions to combat AMR globally.

A multi-faceted approach to improving antimicrobial usage, particularly in LMICs, should include a number of critical activities. Firstly, comprehensive training programs for HCPs to enhance their knowledge and skills in evidence-based prescribing practices and AMR management, starting in universities and continuing post-qualification through continuous professional development. Secondly, effective public education campaigns tailored to address misconceptions about antibiotics and AMR among patients and communities, including culturally and linguistically appropriate messaging. Thirdly, the strengthening of healthcare infrastructure and resource allocation, such as ensuring the availability of diagnostic tools, where pertinent, to guide antibiotic use. Fourthly, the enhancement of patient–provider communication, with an emphasis on shared decision-making and building trust to reduce unnecessary prescribing. Lastly, policy-level interventions, including the implementation of pertinent ASPs and the monitoring of their impact, that are scalable and adaptable to local contexts. It is imperative that countries operate learning health systems and share knowledge and experiences that help health systems to respond to changing environments [135]. This holistic strategy not only addresses the structural and behavioral aspects of antimicrobial usage, but also ensures that interventions are sustainable and effective in diverse healthcare settings.

## 4. Materials and Methods

This scoping review was preregistered on the OSF Database (https://osf.io/2zxhv/?view_only=a78c644446fc4fe88a047093080294d2), accessed on 14 November 2024.

### 4.1. Identifying the Research Question

The research question was designed to explore the key themes of knowledge, attitudes, motivations, and expectations among community members regarding antimicrobial use in PHC settings, particularly in LMICs. Specifically, the review aimed to address the following question: *What are the key themes surrounding patients’ knowledge, attitudes, motivations, and expectations regarding antimicrobial use in community settings?* This question guided the search and selection of relevant literature.

### 4.2. Search Strategy

PubMed, OVID, MEDLINE, CINAHL, relevant antibiotic databases, Scopus, Web of Science, and Clinical Key were selected for conducting searches in relation to the topic. Each of these databases was investigated to guarantee a thorough literature search and to remove any potential bias in the data. The National Library of Medicine created the Medical Subject Headings (MeSH) thesaurus, which is a regulated and hierarchically structured vocabulary. Using this thesaurus, information pertaining to biomedicine and health was indexed, cataloged, and searched across various databases. This allowed for a thorough search and improved the quality of the articles obtained.

### 4.3. Study Selection

The identified studies from the database searches were exported to Covidence, a web-based platform designed to streamline and facilitate the systematic review process, including study screening, data extraction, and collaboration among team members [133]. The PICOS (Population, Intervention, Comparison, Outcome, and Study Design) framework was used to establish the predefined eligibility criteria, ensuring a structured and comprehensive approach to study selection (Table 2).

Two independent reviewers (ET, SC) screened the titles and abstracts of all the studies for relevance based on these criteria. Studies that aligned with the PICOS framework and met the eligibility requirements were included for full-text review. In cases where there were disagreements between the reviewers, a third reviewer (NR) screened for consensus.

### 4.4. Eligibility Criteria

To ensure the relevance and appropriateness of the included studies, the PICOS framework guided the development of strict inclusion and exclusion criteria. These criteria are summarized in Table 3, and the inclusion and exclusion process is summarized in the PRISMA diagram (Figure 1).

### 4.5. Data Collection

Data were collected for this study through a scoping review of the literature. This process involved identifying and selecting studies that met the specific eligibility criteria. Studies were included if they involved community members or patients as the target population and utilized methods such as questionnaires, surveys, or interviews that assessed knowledge, attitudes, motivations, and expectations regarding antimicrobial use.

### 4.6. Data Management

Covidence was used as a tool for managing the screening and data extraction processes [136]. This tool streamlined the study selection and data extraction, and allowed for collaboration between reviewers. The studies identified in the search were imported into Covidence, where two independent reviewers screened the studies for inclusion. Covidence also facilitated the extraction of specific data items from each included study, which were then compiled and organized for further analysis.

### 4.7. Charting the Data [Data Extraction and Analysis]

Data extraction was performed using Covidence. The extracted data included key information related to the study population, study design, interventions, outcomes, and findings regarding knowledge, attitudes, motivations, and expectations surrounding antimicrobial use. Two independent reviewers extracted the data (NR, SC), and discrepancies were resolved by consensus with a third reviewer (ET).

For qualitative studies, the CASP qualitative checklist was used to evaluate the quality of the studies by two independent reviewers (SC, ET) [137]. The CASP tool helped to assess the soundness of the scientific findings and the robustness of the qualitative data reported in the studies based on the clarity of the research aims, the appropriateness of the research methodology, rigor in the data collection process, and the value of the research.

The analysis of the extracted data was conducted using a narrative synthesis approach, as detailed by Popay et al. (2006) and Sukhera (2022) [138,139], which allowed for the identification of common themes and patterns across the selected studies. The themes were grouped into four key areas: ‘*Patient*’, ‘*Provider*’, ‘*Healthcare systems*’, and ‘*Intervention*, *Uptake*, and *Implementation*’. The sub-themes within each of these areas were further categorized, and the number of references supporting each theme was documented.

The narrative synthesis was complemented by a thematic analysis to identify the key drivers of antimicrobial use in PHC settings. To guide the qualitative analysis, the Theory of Planned Behavior (TPB) was employed as a conceptual framework [140]. The thematic analysis was structured to explore how patient beliefs and preferences about antibiotics aligned with the TPB construct of ’attitudes’, how cultural norms, social pressures, and the influence of healthcare providers mapped to ’subjective norms’, and how knowledge, health literacy, and access to healthcare resources were connected to ’perceived behavioral control’. By using the TPB as a lens, the analysis aimed to uncover the behavioral determinants influencing antimicrobial use in LMIC settings, providing a structured and theoretically informed approach to understanding the drivers of antibiotic-related behaviors. Particular attention was paid to studies conducted in LMICs, with a focus on how findings could inform future ASPs in these contexts. Additionally, descriptive statistics were used to summarize the study designs, geographical regions, and types of interventions reported in the included studies.

### 4.8. Item Compilation

The data extraction focused on identifying specific tools or items that measured knowledge, attitudes, motivations, and expectations concerning antimicrobial use. Each study was thoroughly examined to extract relevant measures and any data on the underlying drivers of antimicrobial use. The extracted items were systematically compiled within Covidence to ensure consistent and accurate data management.

### 4.9. Strengths and Limitations of the Study

The study captures antimicrobial use and resistance trends across diverse healthcare systems, providing a global perspective on AMR.A systematic approach using Covidence and the PICOS framework ensures methodological rigor in the study selection and analysis.The study is dominated by research from HICs, with limited contributions from LMICs, despite their significant burden of AMR.The reliance on English-language databases and the exclusion of gray literature may have omitted relevant studies from under-represented regions.

## 5. Conclusions

Overall, we believe that this scoping review has highlighted the critical need for comprehensive, multifactorial, context-specific AMS interventions in PHC settings, particularly in LMICs, to reduce rising AMR rates. By addressing the identified gaps in patient knowledge, provider practices, and healthcare system constraints, key stakeholders can work together towards reducing inappropriate antimicrobial use and combating the global threat of AMR. Addressing such issues is imperative to help deliver on the political declaration of the 79th United Nations General Assembly (UNGA) to reduce human deaths associated with bacterial AMR annually by 10% by 2030.

Our findings reinforce the need for continued research into the drivers of antimicrobial use at the community level and at the PHC level in LMICs, and the importance of public health interventions aimed at fostering informed antibiotic use among community members.

## Figures and Tables

**Figure 1 antibiotics-14-00078-f001:**
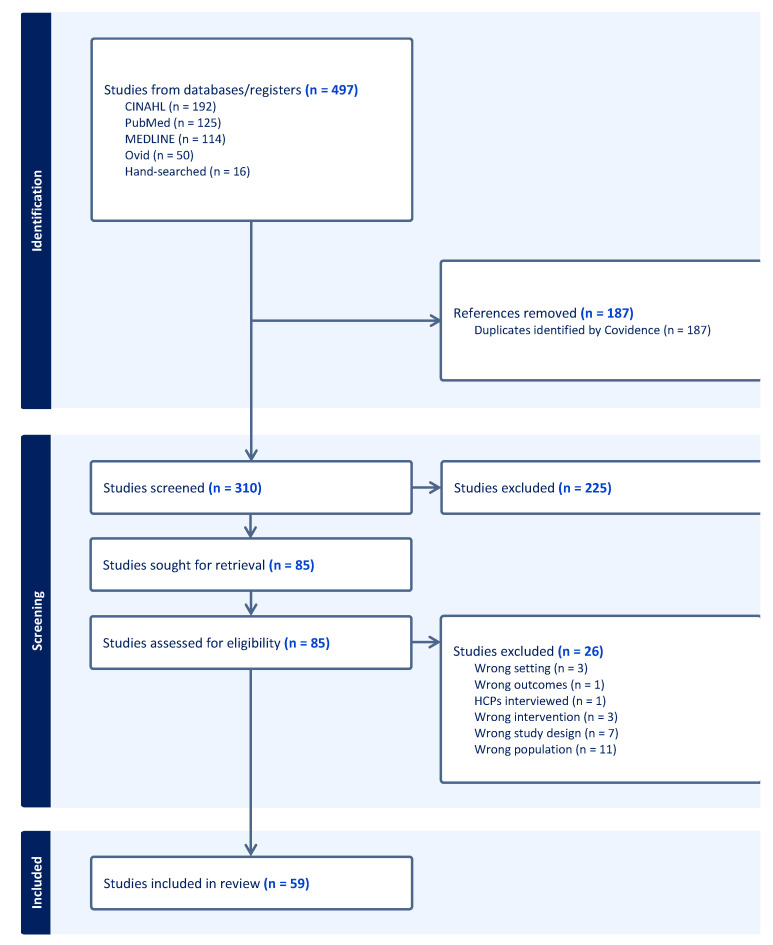
PRISMA flow diagram.

**Table 1 antibiotics-14-00078-t001:** List of key themes and sub-themes.

Themes and Sub-Themes	Description	Number (%) of Sources Featuring Sub-Theme	List of Sources
**Theme 1: Patient**		47 (40.5%)	
Antibiotic usage	Rationalization, patterns, and prevalence of antibiotic utilization	4 (3%)	[68,69,70,71]
Patient–provider dynamics and communication	The impact of patient expectations, educational resources, and antibiotic prescribing decisions	3 (3%)	[72,73,74]
Patient beliefs, expectations, knowledge and understanding, and perceptions of antibiotics, resistance, and illness severity	Maternal decision-making, lay knowledge, and perceptions surrounding antibiotics, illness severity, and antibiotic resistance, to understand patient beliefs, attitudes, and preferences shaping healthcare utilization and antibiotic demand	17 (15%)	[75,76,77,78,79,80,81,82,83,84,85,86,87,88,89,90,91]
Knowledge of infections [general]	Maternal uncertainty and concerns, behaviors, and triggers in the pursuit of seeking reassurance	1 (1%)	[75]
Trust in healthcare providers and shared decision-making and empowerment	Patient trust in healthcare providers, including perceptions of antibiotics’ effectiveness and safety, the impact of pharmaceutical marketing on healthcare provider prescribing behavior, and the importance of patient values and preferences in shared decision-making processes, with an emphasis on patients’ recognition of the significance of careful diagnosis	5 (4%)	[74,92,93,94,95]
Willingness to have blood tests to guide antibiotic use for RTIs	Patients’ willingness to undergo blood tests as a means of informing antibiotic usage for respiratory tract infections	1 (1%)	[87]
Risk perceptions, help-seeking behavior, and treatment preferences	Patient reliance on antibiotics for immediate relief and preferences for prompt treatment, and perceptions of risk and help-seeking behaviors, to understand treatment preferences	2 (2%)	[69,96]
Community attitudes, perceptions, and beliefs, and sociocultural factors	The impact of knowledge and awareness of antibiotic resistance within communities, sociocultural beliefs and practices that influence antibiotic use, diversity in community comprehension of antibiotic resistance and its consequences, and attitudes toward antibiotic use and resistance within communities	10 (9%)	[68,77,80,82,89,92,97,98,99]
Sociodemographic factors and health literacy	How migration, cultural background, sociodemographic factors, educational level, and health literacy influence antibiotic-seeking behavior and knowledge among patients	4 (3%)	[70,76,77,91]
**Theme 2: Provider**		**32 (27.6%)**	
Patient–provider relationship	The dynamics of communication, trust, and decision-making between patients and healthcare providers, including discussions about risks, treatment preferences, and the influence of patient expectations on antibiotic prescribing decisions. Investigating the quality of relationships with HCPs, communication during consultations, and trust in HCPs’ decisions regarding antibiotic prescriptions, while addressing communication challenges and information gaps	13 (11%)	[69,74,75,80,85,88,92,94,96,98,100,101,102]
Healthcare utilization and antibiotic prescribing practices and behaviors	Factors influencing antibiotic prescribing by healthcare providers, including diagnostic uncertainty, clinicians’ concerns and attitudes, awareness of and adherence to prescribing guidelines, and the impact of pharmaceutical marketing practices. Prescribers’ knowledge, skills, and intentions to change behavior, as well as challenges surrounding antibiotic use from the prescribers’ perspective, the appropriateness of prescribing, and the influencing of patient expectations through communication and education	11 (9%)	[71,78,81,85,92,95,100,103,104,105,106]
Antibiotic provision and healthcare delivery in primary healthcare	The role of antibiotics in primary care provision within resource-constrained settings and their impact on healthcare interactions and treatment outcomes. Knowledge and attitudes toward antibiotics and antibiotic resistance among both healthcare providers and patients, and satisfaction with healthcare interactions and treatment outcomes	8 (7%)	[88,97,98,99,101,107,108,109]
**Theme 3: Healthcare system**		**16 (13.8%)**	
Healthcare system challenges and solutions, resource constraints and healthcare delivery, and healthcare system dynamics	Enhancing antibiotic stewardship programs and resources and patient satisfaction with various consultation approaches, and addressing the information needs and preferences of both patients and healthcare providers. Challenges related to accessing healthcare services and antibiotics, enhancing antibiotic stewardship programs and resources, the impact of resource scarcity on healthcare provision, antibiotic availability, prescription patterns, and treatment decisions. Aspects related to follow-up care, healthcare workers’ courtesy, medicine availability, polypharmacy rates, and the presentation, organization, and design of information in healthcare settings. Power dynamics within the healthcare system and the tension between dependency on healthcare providers and autonomy in health decisions. Accessibility to and satisfaction with healthcare services, challenges in PHC services, and the importance of considering population diversity in healthcare interventions	12 (10%)	[68,70,73,79,80,93,96,102,104,105,107,110]
Diagnostic tools and testing	The integration of CRP point-of-care testing into healthcare systems and its influence on treatment-seeking behavior	1 (1%)	[111]
Technological advances and access to care	The feasibility and acceptability of interventions promoting shared decision-making, the influence of CRP point-of-care testing on patients’ perceptions of illness severity, treatment-seeking behavior, and the associated risks of easier access to healthcare advice, potentially leading to antibiotic overuse	3 (3%)	[111,112,113]
**Theme 4: Intervention, uptake, and implementation**		**21 (18.1%)**	
Healthcare utilization and antibiotic prescribing practices	The proportion of patients receiving antibiotics, patterns of antibiotic prescription including delayed prescription, and exploring patients’ confidence in self-care and attitudes toward healthcare utilization	2 (2%)	[93,102]
Uptake of interventions and program components, program effectiveness and outcomes, and public education and awareness	The effectiveness of interventions and public education campaigns in improving knowledge and influencing behavior and attitudes toward antibiotic use among both healthcare providers and patients, while considering variability in the uptake of intervention components and the integration of paper-based versus digital components into daily practice; the intention to change behavior and the impact of interventions on antibiotic prescribing rates, attitudes toward antibiotic use, changes in prescribing practices, patient behaviors post-intervention, and shifts in knowledge levels following educational initiatives; the effectiveness and accessibility of educational materials, emphasizing the importance of clear and plain language in communication. Patients’ awareness and perception of program components, the role of public health messaging, and the impact of public education campaigns on patient behavior and attitudes toward antibiotic use. Misconceptions and lack of understanding regarding antibiotic resistance among the general population, and examining changes in knowledge, perceptions, and attitudes toward antibiotics following educational interventions	15 (13%)	[72,73,85,86,99,106,109,114,115,116,117,118,119,120,121]
Impact of educational intervention on parental antibiotic interest ratings	The effectiveness of video interventions and various messaging strategies in reducing patient demand for antibiotics, and investigating changes in knowledge, perceptions, and attitudes toward antibiotics following educational interventions.	4 (3%)	[115,116,121,122]

NB: HCPs = Healthcare providers, including physicians, pharmacists, and nurses.

**Table 2 antibiotics-14-00078-t002:** Search terms and databases used.

Search terms	“antibiotics”, “antimicrobial use”, “antimicrobial stewardship”, “knowledge”, “attitudes”, “motivations”, “expectations”, “primary healthcare”, “patients”, “community”
Databases	OVID, MEDLINE, PubMed, CINAHL

**Table 3 antibiotics-14-00078-t003:** PICOS framework.

Criteria	Description
Population [P]	Community members or patients seeking care at primary healthcare [PHC] level
Intervention [I]	Exploration of knowledge, attitudes, motivations, and expectations regarding antimicrobial use
Comparison [C]	Not applicable [no specific intervention comparison]
Outcome [O]	Identification of key themes related to knowledge, attitudes, motivations, and expectations regarding antimicrobial use
Study Design [S]	Close-ended questionnaires, surveys, and qualitative studies exploring antimicrobial use

## Data Availability

Additional data are available from the corresponding authors on reasonable request. However, all papers and material have been quoted and are available.

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
