# Peer review of "Knowledge, Attitudes, Motivations, Expectations, and Systemic Factors Regarding Antimicrobial Use Amongst Community Members Seeking Care at the Primary Healthcare Level: A Scoping Review"

_antibiotics, 2025, doi:10.3390/antibiotics14010078_

Round 1

Reviewer 1 Report

Comments and Suggestions for Authors

Thank you for the opportunity to review this manuscript. I hope you find the comments and suggestions below helpful in further strengthening your paper. 

‘Knowledge, Attitudes, Motivations and Expectations Regarding Antimicrobial Use Amongst Community Members Seeking 3 Care at the Primary Health Care Level: A Scoping Review’

Thank you for the opportunity to review this manuscript on knowledge, attitudes, motivations, and expectations of community members regarding antimicrobial use. The review addresses an engaging and timely topic, shedding light on an area thar required further exploration.  

Abstract

The abbreviations in the abstract may be unnecessary, as they are explained in the main text. Removing some of them could improve clarity and flow. Please remove the second full stop after "utilization." The phrase "health literate" seems a little vague and awkward; could you rephrase this sentence?

Introduction

Line 66 – add the year (in 2024).

Lines 112-117 – A recent systematic review by Wojcik et al. (2024) on patient knowledge and understanding of AMR found similar issues in high-income countries, where public campaigns, healthcare, and educational resources are more developed. Although the settings differ, it may be beneficial to acknowledge this review and emphasize that poor patient and public comprehension of the resistance concept (often due to health literacy) is a global issue, not limited to LMICs. This scoping review complements the existing systematic review, and reflecting on its recommendations for developing tailored interventions that address health literacy levels, gender, local context, practices, and other cultural factors influencing antibiotic use would add depth to your discussion.

Wojcik, G., Afseth, J., Fagan, R., Thomson., F. & Ring, N. (2024). Patient and public understanding of antimicrobial resistance: a systematic review and meta-ethnography. JAC-Antimicrobial Resistance. 6(4), 1-16. https://doi.org/10.1093/jacamr/dlae117 

Line 127 - It would strengthen your argument to clearly explain why this review is necessary. Is this area currently underexplored in health research? Additionally, are there existing reviews on this topic that may now be outdated or insufficient? Providing a rationale based on gaps in the literature or the need for updated evidence could enhance the significance of your work.

Results

The PRISMA diagram (Fig 1) is unnecessarily large and has poor resolution. Could it be enhanced?

Line 143 – Can you provide a more detailed description of studies you included? What countries specifically were they from?

Lines 167-215 – Could you link the themes identified in your analysis to the studies from which each theme emerged?

Line 216 Table 1 theme 2 – What’s ‘hcps’ and ‘crp’? Can you explain in full as some readers may not be familiar with those terms? Also, the description of some of the themes and subthemes lacks sufficient depth.

Line 218 – The paragraph in section 2.2 describing the frequency of the sub-themes mentioned in the studies is not very informative. I suggest focusing instead on discussing what these sub-themes entail, and then mention the percentages

Discussion

The Discussion section would benefit from greater comparison with the broader context and existing literature, rather than reiterating the findings. There is a wealth of evidence discussing antimicrobial use in the community in other countries. How do your results align with, refute, or challenge this existing evidence? What novel insights does your review contribute? Also, in your Introduction, you state that this scoping review aims to identify areas for education and behaviour change strategies, but there is no link made to behavioural science and what it can offer. 

Lines 254-6 – What specific language issues are you referring to? The connection you’ve made to surveillance systems in the next sentence is unclear. How does surveillance relate to language? Also, if language is an issue, what interventions would you propose? Did anything emerge from the literature?

Line 276 – What type of behaviours? Can you be more specific?

Line 305 – Can you explain what that multi-faceted approach should look like?

Materials and methods placement after the Results appears unconventional. Typically, most journals require this section to be presented before the Results to provide context for the findings. You may want to confirm whether this arrangement aligns with the specific journal's formatting guidelines or requirements, as it could be a unique policy for that publication.

Line 326 – Did you receive assistance for an academic librarian? If not, how can you be sure your search was comprehensive?

Line 385 - Although quality appraisal is not essential in a scoping review, focusing solely on the quality of qualitative studies seems somewhat limited. It would be beneficial to also assess quantitative studies for the risk of bias to provide a more comprehensive evaluation of the evidence.

Line 381 – How were the discrepancies identified? Did an independent reviewer perform quality assurance of data extraction?

Line 389 – The discussion of data collection reporting appears to be misplaced and would be more appropriately positioned before data extraction. Additionally, elements of data collection have already been mentioned in your search strategy section (line 325) so this is repetition. Could you clarify whether "data collection" here refers to the methods employed in the included studies or the process you used to gather data for your review? This distinction would enhance clarity.

Line 397 – the same applies to data management; this should come a bit earlier.

Line 401 – Was it a blinded approach?

Line 406 – Can you provide a table with the extracted data, including study characteristics?

Lines 413-18 - How did you come up with those themes? Who was involved in the synthesis process?  What steps did you follow? Can you provide some examples in supplementary material of the synthesis/mapping of data?

Lines 427-37 – These are results, not methods, and should be moved to the results section.

Lines 438-443 – This is very brief. Appropriate reporting of the strengths and weaknesses is required.

Additional

·        Some years in the reference list are in bold while others are not. Please ensure consistency.

·        There is inconsistent use of British and American English throughout (e. behaviour and behavior).

Author Response

Quality of English Language
(x) The quality of English does not limit my understanding of the research.
( ) The English could be improved to more clearly express the research

Author comment: Thank you for this.

Is the work a significant contribution to the field?    3/5
Is the work well organized and comprehensively described?    3/5
Is the work scientifically sound and not misleading?    3/5
Are there appropriate and adequate references to related and previous work?    4/5
Is the English used correct and readable?    4/5

Comments and Suggestions for Authors
Thank you for the opportunity to review this manuscript. I hope you find the comments and suggestions below helpful in further strengthening your paper. 

Thank you for the opportunity to review this manuscript on knowledge, attitudes, motivations, and expectations of community members regarding antimicrobial use. The review addresses an engaging and timely topic, shedding light on an area thar required further exploration.  

Author comments: Thank you for your kind words and comments. We hope we have satisfactorily addressed these.

A) Abstract

The abbreviations in the abstract may be unnecessary, as they are explained in the main text. Removing some of them could improve clarity and flow. Please remove the second full stop after "utilization." The phrase "health literate" seems a little vague and awkward; could you rephrase this sentence?

Author response: Thank you for these. We are obliged to include abbreviations where possible. However, we have removed the full stop. The phrase health literate has also been revised. We trust this is now acceptable  

B) Introduction

i) Line 66 – add the year (in 2024).

Author response: Now added in 

ii) Lines 112-117 – A recent systematic review by Wojcik et al. (2024) on patient knowledge and understanding of AMR found similar issues in high-income countries, where public campaigns, healthcare, and educational resources are more developed. Although the settings differ, it may be beneficial to acknowledge this review and emphasize that poor patient and public comprehension of the resistance concept (often due to health literacy) is a global issue, not limited to LMICs. This scoping review complements the existing systematic review, and reflecting on its recommendations for developing tailored interventions that address health literacy levels, gender, local context, practices, and other cultural factors influencing antibiotic use would add depth to your discussion.
Wojcik, G., Afseth, J., Fagan, R., Thomson., F. & Ring, N. (2024). Patient and public understanding of antimicrobial resistance: a systematic review and meta-ethnography. JAC-Antimicrobial Resistance. 6(4), 1-16. https://doi.org/10.1093/jacamr/dlae117 

Author response: The introduction and discussion sections have now been updated to include this study and reference to the WHO definition of health literacy. We hope this is now OK.

iii) Line 127 - It would strengthen your argument to clearly explain why this review is necessary. Is this area currently underexplored in health research? Additionally, are there existing reviews on this topic that may now be outdated or insufficient? Providing a rationale based on gaps in the literature or the need for updated evidence could enhance the significance of your work.

Author response:  Thank you for your comments. We emphasize that this scoping review builds on existing evidence by specifically addressing critical gaps in (1) understanding how community members in LMICs perceive and engage with AMR; (2) the role of health literacy, cultural factors, and local healthcare practices in influencing antimicrobial use; and (3) the development of tailored, context-specific interventions to address these issues. Existing reviews often focus on high-income countries or address AMR through a narrow lens, leaving these important questions underexplored both generally and especially in LMICs. Addressing inappropriate use of antibiotics in ambulatory care in LMICs is critical to reduce AMR in line with the UNGA Goals especially as antibiotic utilization in LMICs can account for up to 95% of total antibiotic use in humans and rates of AMR are considerably higher in LMICs than high income countries. We hope this provides enough of a rationale. 

C) Results

i) The PRISMA diagram (Fig 1) is unnecessarily large and has poor resolution. Could it be enhanced?

Author response: The diagram has been re-inserted in high resolution format. We trust this is now OK. 

ii) Line 143 – Can you provide a more detailed description of studies you included? What countries specifically were they from?

Author response: The countries that the studies were from have been included in lines 171-177, i.e.  the United Kingdom (UK) accounted for the highest number of studies (15), followed by the United States (6), Spain (5), and broader European contributions (4). Canada and Australia each contributed 3 studies. Additional single-study contributions came from countries such as France, Brazil, Sudan, Saudi Arabia, Nigeria, Finland, Iceland, Italy, Germany, Malawi, South Africa, Singapore, China, Egypt, Norway, Myanmar, Thailand, Ireland, Malaysia, Serbia, and India. We have also documented this further in the Supplementary Material summarizing key aspects of each study including the countries involved in the research. We hope this is now acceptable.

iii) Lines 167-215 – Could you link the themes identified in your analysis to the studies from which each theme emerged?

Author response:  Thank you. Table 1 lists the themes and subthemes including the sources/studies.

iv) Line 216 Table 1 theme 2 – What’s ‘hcps’ and ‘crp’? Can you explain in full as some readers may not be familiar with those terms? Also, the description of some of the themes and subthemes lacks sufficient depth.

Author response:  Thank you. Line 113/114 (tract changes) states “healthcare professionals (HCPs)”, which has been capitalized and examples of HCPs added in. Line 230/231 under healthcare systems: “Additionally, this theme covers the integration of diagnostic tools including C-Reactive Protein (CRP) point-of-care testing”. Descriptions of the themes and subthemes have now been updated.

v) Line 218 – The paragraph in section 2.2 describing the frequency of the sub-themes mentioned in the studies is not very informative. I suggest focusing instead on discussing what these sub-themes entail, and then mention the percentages

Author response:  Thank you. Section 2.2. has been augmented with discussion of each sub-theme. We hope this is now OK.

D) Discussion

i) The Discussion section would benefit from greater comparison with the broader context and existing literature, rather than reiterating the findings. There is a wealth of evidence discussing antimicrobial use in the community in other countries. How do your results align with, refute, or challenge this existing evidence? What novel insights does your review contribute? Also, in your Introduction, you state that this scoping review aims to identify areas for education and behaviour change strategies, but there is no link made to behavioural science and what it can offer. 

Author response: Thank you. The discussion has been updated to align the findings from the 4 key themes identified in this scoping review with the gaps in the current literature stated in the introduction. We trust this is now acceptable.  

ii) Lines 254-6 – What specific language issues are you referring to? The connection you’ve made to surveillance systems in the next sentence is unclear. How does surveillance relate to language? Also, if language is an issue, what interventions would you propose? Did anything emerge from the literature?

Author response:  Sentences on language issues and surveillance systems have now been updated, and also emphasized in the conclusions. We trust this is now OK.

iii) Line 276 – What type of behaviours? Can you be more specific?

Author response: Thank you. We have expanded on the examples of what behaviors mean here in relation to the 5th sub-theme, and hope this is now OK. 

iv) Line 305 – Can you explain what that multi-faceted approach should look like?

Author response: The text has been updated to include the following in LMICs to illustrate this point  - particularly in LMICs, should include: (1) comprehensive training programs for HCPs to enhance their knowledge and skills in evidence-based prescribing practices and AMR management; (2) effective public education campaigns tailored to address misconceptions about antibiotics and AMR among patients and communities, including culturally and linguistically appropriate messaging; (3) strengthening healthcare infrastructure and resource allocation, such as ensuring the availability of diagnostic tools to guide antibiotic use; (4) enhancing patient-provider communication, with an emphasis on shared decision-making and building trust to reduce unnecessary prescribing; and (5) policy-level interventions, such as implementing and monitoring ASPs that are scalable and adaptable to local contexts.  We hope this adequately addresses this concern. 

E) Materials and Methods

i) Materials and methods placement after the Results appears unconventional. Typically, most journals require this section to be presented before the Results to provide context for the findings. You may want to confirm whether this arrangement aligns with the specific journal's formatting guidelines or requirements, as it could be a unique policy for that publication.

Author response: Thank you for this. However, as you are no doubt aware of the Template that the Journal asks us to use for submissions - this is a requirement of the Journal. 

ii) Line 326 – Did you receive assistance for an academic librarian? If not, how can you be sure your search was comprehensive?

Author response: Thank you for this comment. We did not receive direct assistance from an academic librarian for this review. However, the authors include Cochrane Systematic Review experts, skilled in developing search terms and strategies for identifying articles in databases and we adopted a systematic and comprehensive approach to ensure the rigor of the search process. Our search strategy was carefully developed and implemented using key databases, including OVID Medline, PubMed, and CINAHL, which are widely recognized as primary sources for health and medical literature. Boolean operators and MeSH terms were systematically combined to optimize search precision and relevance, with search terms tailored to capture studies on antimicrobial use, community perceptions, and related behaviors. Furthermore, the study selection process was enhanced by Covidence, a web-based platform designed to facilitate systematic reviews, ensuring accuracy in screening and data extraction. The use of the PICOS framework further strengthened the search and study selection process by applying predefined eligibility criteria. To minimize bias, two independent reviewers screened the studies, with a third reviewer resolving any disagreements.
Consequently, whilst direct input from an academic librarian was not sought, we believe the rigorous methodology and use of validated frameworks and tools that we used reflects a high standard of comprehensiveness and reliability. We trust this adequately addresses your concerns.   

iii) Line 385 - Although quality appraisal is not essential in a scoping review, focusing solely on the quality of qualitative studies seems somewhat limited. It would be beneficial to also assess quantitative studies for the risk of bias to provide a more comprehensive evaluation of the evidence.

Author response:  We sought to address potential risks of bias across all included studies, regardless of methodology and used Covidence for all studies. For a comprehensive evaluation, we captured information such as potential conflicts of interest, funding sources, and reported strengths and limitations for both qualitative and quantitative studies. In this way we were able to identify and note any potential biases in the evidence. Whilst we did not formally assess the risk of bias in quantitative studies, we believe this captured information offers valuable insights into the reliability and validity of the findings. We trust you agree with us.

iv) Line 381 – How were the discrepancies identified? Did an independent reviewer perform quality assurance of data extraction?

Author response: The data extraction process was conducted by two independent reviewers. In cases where discrepancies were identified between their extractions, a third reviewer, who had not been involved in the initial extraction process, was brought in to resolve the issue. The third reviewer reviewed the discrepancies and worked with the two original reviewers to reach a consensus. If necessary, further discussion was held to clarify any points of disagreement and ensure consistency and accuracy in the data. This approach helped maintain the integrity of the data extraction process and ensured that any discrepancies were thoroughly addressed.

v) Line 389 – The discussion of data collection reporting appears to be misplaced and would be more appropriately positioned before data extraction. Additionally, elements of data collection have already been mentioned in your search strategy section (line 325) so this is repetition. Could you clarify whether "data collection" here refers to the methods employed in the included studies or the process you used to gather data for your review? This distinction would enhance clarity. Line 397 – the same applies to data management; this should come a bit earlier.

Author response:  The text has been moved to before data extraction and the data collection updated so there is no repetition with search strategy. Data collection refers to the process used to gather data for the review, updated under data collection. In addition, Data management has been moved up before charting the data. We hope this is now acceptable.

vi) Line 401 – Was it a blinded approach?

Author response:  The approach used in this review was not blinded. Reviewers were not blinded to the authorship, journal, or other identifying details of the studies during the screening or data extraction process. We mitigated potential biases by independent screening and data extraction by two reviewers. Discrepancies were resolved by a third reviewer who was not involved in the initial extraction process. Additionally, the use of predefined eligibility criteria and frameworks, such as PICOS, ensured consistency and objectivity in the study selection and data collection process. We trust this addresses any concerns you may have with our approach.

vii) Line 406 – Can you provide a table with the extracted data, including study characteristics?

Author response:  Thank you - this has been added as supplementary material

viii) Lines 413-18 - How did you come up with those themes? Who was involved in the synthesis process?  What steps did you follow? Can you provide some examples in supplementary material of the synthesis/mapping of data?

Author response:   The themes and sub-themes presented in the study were derived through a structured synthesis process. All authors were involved (NR, SC, ET, BG, NS, JCM). We started by exporting charted data from Covidence, which included key findings and characteristics from the included studies. These findings were grouped into broader themes, such as "Patient," "Provider," "Healthcare System," and "Intervention, Uptake, and Implementation," by identifying recurring patterns and similarities. Within these broad themes, further analysis was conducted to identify sub-themes, ensuring that all relevant nuances in the data were captured.

ix) Lines 427-37 – These are results, not methods, and should be moved to the results section.

Author response:  This has now been updated.

x) Lines 438-443 – This is very brief. Appropriate reporting of the strengths and weaknesses is required.

Author response:  Thank you. Additional text on the strengths and limitations has been inserted before the conclusion (4.9), and trust this is now acceptable, 

F) Additional

i) Some years in the reference list are in bold while others are not. Please ensure consistency.

Author response:  We will work with the Journal to update the reference style where needed if and when the paper is accepted for publication. We hope this is OK.

ii) There is inconsistent use of British and American English throughout (e. behaviour and behavior).

Author response:  American English has been used throughout. 

Reviewer 2 Report

Comments and Suggestions for Authors

The manuscript (antibiotics-3383122) highlights a crucial public health issue, i.e . antimicrobial resistance seen while antimicrobial usage at community level. Although , the manuscript covers many important areas but there is still lot of portions that need significant imporvements and justification. The manscuript should be revised for consistent use of terminology and avoid usage of repeated sentences. The below points should be taken into consideration before resubmission:

1.     Authors should highlight the contribution of this review and what gaps does the manuscript focusses upon.

2.     Author should consider including detailed table summarizing search terms and databases for improving clarity to readers.

3.     Explain the criteria for exclusion studies, specifically the ones which are considered as irrelevant during the screening.

4.     Author should provide detailed description on geographical representation of studies included in this review.

5.     Provide more details aout critical analysis with respect to the research gaps in existing studies, specifically how cultural and systemic factors in low and middle income countries effects AMR.

6.     Authors should consider involving specific recommedations for applying the known interventions and approaches taken when resources are restricted.

Author Response

Quality of English Language
(x) The quality of English does not limit my understanding of the research.
( ) The English could be improved to more clearly express the research.

Author comments: Thank you for this

Is the work a significant contribution to the field?    3/5
Is the work well organized and comprehensively described?    2/5
Is the work scientifically sound and not misleading?    3/5
Are there appropriate and adequate references to related and previous work?    2/5
Is the English used correct and readable?    4/5

Comments and Suggestions for Authors

The manuscript (antibiotics-3383122) highlights a crucial public health issue, i.e . antimicrobial resistance seen while antimicrobial usage at community level. 

Author response:  Thank you for this – appreciated!

Although, the manuscript covers many important areas but there is still lot of portions that need significant imporvements and justification. The manscuript should be revised for consistent use of terminology and avoid usage of repeated sentences. The below points should be taken into consideration before resubmission:

Author response: Thank you for your review. We hope the revised manuscript is now acceptable. 

1.     Authors should highlight the contribution of this review and what gaps does the manuscript focusses upon.

Author response:  We have updated  the introduction to emphasize the critical gaps addressed in this scoping review, and trust this is now acceptable.

2.     Author should consider including detailed table summarizing search terms and databases for improving clarity to readers.

Author response:  This has been included under the search strategy. We trust this is now OK.

3.     Explain the criteria for exclusion studies, specifically the ones which are considered as irrelevant during the screening.

Author response:  Thank you. The first paragraph under the results contains the following: “As shown in Figure 1, our search identified 497 records, which included 192 from CINAHL, 125 from PubMed, 114 using MEDLINE, 50 using Ovid and a further 16 from unspecified sources. In total, 187 duplicate records were removed using the Covidence Software. The titles and abstracts of the remaining 310 records were screened for relevance, with 225 records subsequently excluded for not meeting the inclusion criteria. Full-text eligibility screening was performed on the remaining 85 records, and a further 26 records were subsequently excluded. Of these, 11 had a different patient population, 7 used different study designs, 3 assessed different interventions, and 3 were excluded for different study settings. One study interviewed HCPs, and one did not report the outcomes of interest. In total, 59 articles were included in the review.” We trust this is now OK.

4.     Author should provide detailed description on geographical representation of studies included in this review.

Author response:  Thank you. The results section includes the geographical representation of the studies: “The studies included in the analysis originated from a diverse set of countries, highlighting a broad geographical representation. Specifically, the United Kingdom (UK) accounted for the highest number of studies (15), followed by the United States (6), Spain (5), and broader European contributions (4). Canada and Australia each contributed 3 studies. Additional single-study contributions came from countries such as France, Brazil, Sudan, Saudi Arabia, Nigeria, Finland, Iceland, Italy, Germany, Malawi, South Africa, Singapore, China, Egypt, Norway, Myanmar, Thailand, Ireland, Malaysia, Serbia, and India. This diversity underscores the global scope and multidisciplinary nature of the research conducted.”

5.     Provide more details about critical analysis with respect to the research gaps in existing studies, specifically how cultural and systemic factors in low and middle income countries effects AMR.

Author response:  Thank you. Text has now been added into the discussion, and hope this is now OK.

6.     Authors should consider involving specific recommendations for applying the known interventions and approaches taken when resources are restricted.

Author response: Thank you - recommendations have now been added to the discussion, and we trust this is now acceptable.

Reviewer 3 Report

Comments and Suggestions for Authors

1. The title of the study does not appropriately describe the aim and overall findings of the study. At the beginning, I perceived that the authors would describe the overall themes of knowledge, attitude, motive and expectation. However, another key themes not related to knowledge, attitude, and motive and expectation were also described.

2. Introduction could be improved by making it more concise and please consider to include another systematic review about KAP in antibiotics use in primary care.

3. Methodology: what kind of theoretical framework does the authors refer to? Authors are strongly encouraged to provide the theory of behaviour as the fondartion to analysis the findings. 

4. Methodology: how do the authors define the Primary Care owing that the scope of primary care vary across countries.

5. Results: The themes presented are lack with interconnection association and this could be related to the lack of theoretical framework used by the authors.

6. Lines 427-443 are inappropriately placed in the methodology. Please consider to move it to introduction.

7. Lines 237-234 in the discussion part repeat the results.

Author Response

Quality of English Language
(x) The quality of English does not limit my understanding of the research.
( ) The English could be improved to more clearly express the research.

Author comments: Thank you for this

Is the work a significant contribution to the field?    2/5
Is the work well organized and comprehensively described?    4/5
Is the work scientifically sound and not misleading?    4/5
Are there appropriate and adequate references to related and previous work?    2/5
Is the English used correct and readable?    4/5

Comments and Suggestions for Authors

1. The title of the study does not appropriately describe the aim and overall findings of the study. At the beginning, I perceived that the authors would describe the overall themes of knowledge, attitude, motive and expectation. However, another key themes not related to knowledge, attitude, and motive and expectation were also described.

Author response: Thank you. The title has been changed to “Knowledge, Attitudes, Motivations, Expectations and Systemic Factors Regarding Antimicrobial Use Amongst Community Members Seeking Care at the Primary Health Care Level: A Scoping Review”. We trust this is now OK

2. Introduction could be improved by making it more concise and please consider to include another systematic review about KAP in antibiotics use in primary care.

Author response:  Thank you. We have revised the introduction where possible. However – e had to accommodate comments from other Reviewers as well. We hope this is now OK.

3. Methodology: what kind of theoretical framework does the authors refer to? Authors are strongly encouraged to provide the theory of behaviour as the foundation to analysis the findings. 

Author response:  No theoretical framework was used in this scoping review, which used an extensive synthesis of approaches found in existing literature to understand and/or explain what factors are important to address the stated aims. However, the analyses was guided by the theory of planned behaviour and text has been added. We trust this is acceptable.

4. Methodology: how do the authors define the Primary Care owing that the scope of primary care vary across countries.

Author response: Thank you. Primary care in this review is defined as the first point of contact within the healthcare system, where individuals receive accessible and comprehensive healthcare services. This encompasses care provided by general practitioners, nurses, pharmacists, and other non-specialist healthcare workers in formal settings such as clinics and community health centers. aligning with the WHO's framework for community-based healthcare.

5. Results: The themes presented are lack with interconnection association and this could be related to the lack of theoretical framework used by the authors.

Author response: As above, the analyses and themes were guided by the TPB, and text has been added to the manuscript.  

6. Lines 427-443 are inappropriately placed in the methodology. Please consider to move it to introduction.

Author response:  Thank you, this has been updated and paragraph moved to discussion as appropriate. We hope this is now OK.  

7. Lines 237-234 in the discussion part repeat the results.

Author response:  Thank you, this has been updated in the discussion.

Round 2

Reviewer 2 Report

Comments and Suggestions for Authors

The manuscript can be considered for publication in its current form.